# DDX21 Controls Cell Cycle Progression and Autophagy in Pancreatic Cancer Cells

**DOI:** 10.3390/cancers17040570

**Published:** 2025-02-07

**Authors:** Adriana Leccese, Veronica Ruta, Valentina Panzeri, Fabia Attili, Cristiano Spada, Valentina Cianfanelli, Claudio Sette

**Affiliations:** 1Department of Neuroscience, Section of Human Anatomy, Catholic University of the Sacred Heart, 00168 Rome, Italy; adriana.leccese@unicatt.it (A.L.); veronica.ruta@unicatt.it (V.R.); valentina.panzeri@unicatt.it (V.P.); 2Fondazione Policlinico A. Gemelli IRCCS, 00168 Rome, Italy; fabia.attili@policlinicogemelli.it (F.A.); cristiano.spada@unicatt.it (C.S.); valentina.cianfanelli@uniroma3.it (V.C.); 3Department of Translational Medicine & Surgery, Catholic University of the Sacred Heart, 00168 Rome, Italy; 4Department of Science, University “Roma TRE”, 00146 Rome, Italy

**Keywords:** pancreatic ductal adenocarcinoma, autophagy, cell cycle

## Abstract

Pancreatic ductal adenocarcinoma (PDAC) is a highly aggressive disease, with a 5-year survival rate < 10%. Current therapies are poorly effective, thus urging the identification of new therapeutic approaches to face this lethal cancer. The RNA helicase DDX21 was recently shown to be upregulated and to associate with poor prognosis in PDAC. Our study shows that DDX21 is further upregulated in liver metastasis with respect to the primary PDAC lesions, and that depletion of DDX21 triggers autophagy while perturbing the cell cycle progression of PDAC. Together, our data support the oncogenic function of DDX21 in PDAC cells and uncover biological processes and pathways modulated by this RNA helicase.

## 1. Introduction

Pancreatic ductal adenocarcinoma (PDAC) is among the most lethal human cancers, with an overall 5-year survival of ~10% [1]. Surgery followed by adjuvant multidrug chemotherapy remains the most efficacious therapy for PDAC. However, the majority of cases are diagnosed late, with locally advanced or metastatic disease, which limits their eligibility to surgery [1,2,3]. Moreover, most PDACs display intrinsic or acquired resistance to chemotherapy, further reducing the efficacy of the current therapeutic options. The high heterogeneity of PDAC is thought to contribute to the poor response to chemotherapy. Genome-wide transcriptomic studies have clearly shown the presence of two cancer cell-intrinsic PDAC subtypes, named classical and basal-like or squamous [4]. The classical subtype maintains traits of glandular differentiation, expression of endoderm lineage transcription factors, like GATA6, and is characterized by relatively better prognosis and improved response to mFOLFIRINOX (modified 5′-fluorouracil, leucovorin, irinotecan, oxaliplatin). The basal-like/squamous subtype is characterized by a worse prognosis and by the expression of epithelial-to-mesenchymal transition (EMT) factors [2,4]. However, most PDACs comprise a mixture of cells of both subtypes, which can also convert into each other upon exposure to specific environmental cues [2]. Moreover, the identification of these molecular traits has not yet provided robust and exploitable therapeutic targets. Thus, the identification of biomarkers that are capable to identify the tumor at an early stage and/or the development of more efficacious therapies represent clinical priorities for this disease [1,2,3].

The functional dysregulation of RNA-binding proteins and other proteins involved in RNA metabolism is emerging as a hallmark of PDAC [5]. For instance, it was shown that the splicing factor Serine And Arginine Rich Splicing Factor 1 (SRSF1) contributes to pancreatic inflammation at the onset of tumorigenesis [6] and to drug resistance in established PDAC cells [7]. Furthermore, a specific splicing signature allows the identification of PDAC molecular subtypes and it is significantly associated with prognosis in patients [8]. In particular, the splicing factor Quaking (QKI) was shown to promote the basal-like splicing signature and a plastic, pro-mesenchymal phenotype of PDAC cells [8]. Another RNA-processing factor of relevance for PDAC is DDX21, a member of the DExD-box RNA helicase family characterized by the Asp-Glu-Ala-Asp (DEAD) motif [9]. DDX21 is involved in RNA synthesis, processing and editing, particularly of RNA polymerase I encoded ribosomal genes [10,11,12]. However, recent data also indicated that DDX21 can translocate from nucleoli to nucleoplasm in response to environmental cues, thus impacting specific splicing events [13]. An aberrant expression of DDX21 was observed in several cancers, with somewhat opposite roles depending on the specific context examined. Indeed, while DDX21 plays an oncogenic role in colorectal [14], gastric [15] and other cancers [16], downregulation of its expression correlates with dismal prognosis in breast cancer [17] and kidney renal papillary cell carcinoma [16]. In the context of PDAC, a recent study suggested a pro-oncogenic role for DDX21, as its expression is augmented in tumors with respect to normal pancreatic tissues and was associated with worse prognosis [18]. At the cellular level, depletion of DDX21 in the PDAC cell line MiaPaCa-2 reduced several oncogenic features [18]. Nevertheless, the genes and pathways regulated by this helicase in PDAC cells were not investigated. Herein, we have further explored the role of DDX21 by analyzing its expression and function in PDAC biopsies and cells. Our transcriptomic and functional analyses uncover a new role of DDX21 in the regulation of autophagy in PDAC cells. Furthermore, DDX21 supports cell proliferation by promoting the G1-S transition of the cell cycle through the expression of Cyclin D3 (CCND3) and its cognate Cyclin Dependent Kinase 6 (CDK6). Together, our study further supports the pro-oncogenic activity of DDX21 in this highly aggressive and lethal tumor type.

## 2. Materials and Methods

### 2.1. Cell Culture, Transfection, Western Blot

HPAF-II, Capan-1 and PANC-1 cells were grown in RPMI 1640 (Euroclone, Pero, MI, Italy); MiaPaCa-2 and AsPC-1 cells were grown in DMEM (Sigma-Aldrich, St. Louis, MO, USA). All media were supplemented with 10% fetal bovine serum (FBS, Thermo Fisher Scientific, Waltham, MA, USA), Non-Essential Amino Acids 100× (Thermo Fisher Scientific, Waltham, MA, USA), gentamycin (Biowest, Nuaillè, France), penicillin and streptomycin (Biowest, Nuaillè, France). PDAC biopsies were collected upon informed consent from patients undergoing endoscopic ultrasound guided tissue acquisition (EUS-TA) for diagnostic purpose at Fondazione Policlinico Universitario A. Gemelli IRCCS, Rome, Italy, from May 2023 to October 2023. The protocol (ID:4121) was approved by the Institutional Review Board and the procedures were conducted in accordance with the Helsinki Declaration.

For RNA interference, cells were transfected with the Darmachon smartpool siRNAs using Lipofectamine RNAiMAX (Thermo Fisher Scientific, Waltham, MA, USA) and harvested after 48 h for protein and RNA analyses. For western blot analyses, cell pellets were resuspended in RIPA buffer (Tris-HCl 50 mM, NP40 1%, NaCl 150 mM, Na-Deoxycholate 0.5%, EDTA 2 mM, SDS 0.1%) supplemented with 2 mM Na-orthovanadate, 0.5 mM sodium fluoride, 1 mM dithiothreitol and Protease-Inhibitor Cocktail (Sigma-Aldrich, St. Louis, MO, USA). Extracts were incubated for 10 min on ice, sonicated for 5 s and centrifuged for 10 min at 15,000× *g*, 4 °C. Supernatants were diluted in SDS-PAGE sample buffer, boiled for 10 min and analyzed as described. Primary antibodies: anti-DDX21 (Santa Cruz Biotechnology, Dallas, TX, USA, sc-376953), anti-LC3 XP (Cell Signaling Technology, Danvers, MA, USA, CS-3868), anti-ATG5 (Cell Signaling, Danvers, MA, USA, 2630S), anti-HSP90 (Santa Cruz Biotechnology, Dallas, TX, USA, sc-13119), anti-Actin (Santa Cruz Biotechnology, Dallas, TX, USA, sc-47778).

### 2.2. RNA Extraction and RT-PCR Analyses

Total RNA was extracted using the Geneaid Minikit (Geneaid, Xizhi District, New Taipei City, Taiwan). After digestion with DNase, 1 mg of RNA was reverse-transcribed using M-MLV reverse transcriptase and random hexamers (both from Promega Corporation, Madison, WI, USA). Real-time quantitative PCRs (qPCR) were performed using the SYBR Green I Master and the LightCycler 480 System (Roche, Basel, Switzerland) [8,19]. All primers used are listed in Appendix A.

### 2.3. RNA-Seq Analysis

Total RNA was extracted, and DNase treated using the RNeasy mini kit (QIAGEN, Hilden, Germany) from PANC-1 cells transfected for 48 h with control (*n* = 3) or DDX21-targeting (*n* = 3) siRNAs. PolyA plus RNA sequencing (RNA-seq) libraries were constructed and sequenced using a 150 bp paired-end format on an Illumina NovaSeq 6000 at high depth (>100 million reads/sample) [8,20,21]. Repartition of raw quality reads was performed using FastQC v0.11.2 and reads were mapped using STAR v2.7.9a on the human hg38 genome assembly. Inner distance size estimation, gene body coverage, chromosome coverage and strand specificity of aligned reads were performed using Picard-Tools v3.0.0, Samtools v1.13 and RSeQC v4.0.0. Read count was performed using featureCounts v2.0.3. Gene expression estimation was performed with DESeq2 using Human FAST DB v2022 1 annotations [8,20,21]. Differential expressed genes were considered statistically significant for *p*-values ≤ 0.05 and fold-changes ≥ 1.5. Splicing analyses were performed considering only exon reads and flanking exon–exon junction reads (“EXON” analysis) in order to detect new potential alternative events and known patterns (“PATTERN” analysis) using the Human FAST DB v2022_1 splicing patterns annotation [8,20,21]. Results were considered statistically significant for *p*-values < 0.05 and fold-changes ≥ 1.5 for “PATTERN” analysis and *p*-values < 0.01 and fold-changes ≥ 2 for “EXON” analysis. Correlation of RBP expression with target genes in TCGA Firehose Legacy dataset was performed with the cBioPortal database (https://www.cbioportal.org/, accessed on 2 December 2024).

### 2.4. Cell Proliferation, Viability and Clonogenic Assays

Cells were seeded in a 96-well plate and imaged at 10× magnification in an IncuCyte SX5 Live-content imaging system (Essen Bioscience, Ann Arbor, MI, USA) at 37 °C with 5% CO_2_. Cells were labeled with Nuclight Rapid NIR (Sartorius, Göttingen, Germany) and images were acquired every 12 h for 6 days (four images/well) and analyzed using the IncuCyte Cell-by-Cell software vSX5 to detect and quantify live cells. For mFOLFIRINOX treatment, a mixed solution of fluorouracil (0.4 μM, 1.3 μM, 4 μM), oxaliplatin (0.4 μM, 1.3 μM, 4 μM) and irinotecan (0.2 μM, 0.7 μM, 2 μM) was dispensed. Gemcitabine was administered at 10 nM, 15 nM and 30 nM concentration. All drugs were purchased from MedChem Express (Monmouth Junction, NJ, USA).

Colony-forming assays were performed by seeding 700 PANC-1 cells silenced or not for DDX21 in 6-multiwell. After 10 days, cells were fixed for 30 min with glutaraldehyde 6.0% (*v*/*v*)/crystal violet 0.5% (*w*/*v*) solution and colonies with *n* > 50 cells were counted.

### 2.5. Immunofluorescence Assays

PANC-1 cells were fixed with 4% paraformaldehyde and permeabilized with 0.5% Triton X-100. After 1 hour (h) at room temperature (RT) in blocking buffer (PBS, 5% BSA, 3% horse serum), they were incubated overnight with anti-DDX21 antibody (Protein Tech, Rosemont, IL, USA 10528-1-AP) at 1:500 dilution in blocking buffer. After rinsing, secondary anti-mouse Alexa Fluor 594/anti-rabbit Alexa Fluor 488 (1:400; Thermo Fisher Scientific, Waltham, MA, USA) antibodies diluted in PBS supplemented with 1% BSA were incubated for 1 h at 37 °C. Nuclei were counterstained with Hoechst 33342. Images were minimally processed with Adobe Photoshop for composing panels. For the analysis of LC3 dots, control and DDX21-depleted PANC-1 cells were treated or not with 50 nM Bafilomicin A1 (BAF) (Santa Cruz Biotechology, Dallas, TX, USA, cs-201550A) for 2 h. After the treatment, PANC-1 cells were fixed with 4% formaldehyde for 10 min at RT and with cold 100% Met-OH for 10 min at −20 °C. After 1 h of blocking in PBS supplemented with 3% goat serum at RT, cells were incubated for 2 h, at RT, with anti-LC3 XP (Cell Signaling Technology, Danvers, MA, USA, CS-3868) at 1:150 dilution. After rinsing, cells were stained for 45 min with Alexa Fluor anti-rabbit secondary 488 antibody (1:300). Nuclei were counterstained with Hoechst (1:2000 in PBS) for 10 min. Immunofluorescence images were captured using a 63× objective of the Zeiss Axiophot fluorescence microscope. Image analysis for LC3 dot quantification was performed using ImageJ software, https://imagej.net/ij/index.html. Images were processed to equally remove the background without affecting the original signal.

### 2.6. Cell Cycle Analysis

PANC-1 cells were silenced with siDDX21 and ctrl siRNAs and, after 48 h, they were prepared for the cell cycle analysis [22,23]. Cell cycle was evaluated by flow cytometry using single staining with 7-aminoactinomycin D (7-AAD; 2.5 g/mL, Biotium, Fremont, CA, USA) or double staining with 7-AAD and anti-BrdU antibody (0.125 μg for each sample; Becton Dickinson, Franklin Lakes, NJ, USA), in the presence of ribonuclease A (1 μg/mL), as previously described [22,23]. Cell cycle phases were set using asynchronized control samples. A total of 10,000 events were counted using the BD FACSCanto flow cytometer and analyzed using FLOWJO v.10 software (both from Becton Dickinson, Franklin Lakes, NJ, USA).

### 2.7. Statistical Analyses

All experiments were carried out in biological triplicates, unless otherwise specified, and *p*-values were calculated using GraphPad Prism 7.0a software. For real-time analysis and western blot analyses, statistics were performed using Student’s *t*-test (two tails). The quantitative data in the graphs are expressed as mean ± standard deviation (SD). Statistical analyses for the LC3 count were performed using the Two-Way ANOVA. Differences showing a *p*-value ≤ 0.05 were regarded as statistically significant.

## 3. Results

### 3.1. Expression of DDX21 Correlates with That of Genes Involved in Transcription, Metabolism and Cell Cycle Regulation in PDAC

Analysis of transcriptomic data from The Cancer Genome Atlas (TCGA) Firehose legacy project indicated that *DDX21* transcript expression is elevated in 5% of PDAC samples (Figure 1A). Coherently, the DDX21 protein was also found to be upregulated in 5% of PDAC samples from the Clinical Proteomic Tumor Analysis Consortium (CPTAC [24]; Figure 1B). As previously reported [16,18], higher expression of DDX21 was significantly associated with worse disease-free survival (DFS) and overall survival (OS) in PDAC patients (Figure 1C), suggesting an oncogenic impact of this helicase in pancreatic cells. Next, we asked whether a high *DDX21* expression was associated with a specific gene signature in PDAC. To this end, we selected the 100 genes with the highest expression correlation score from TCGA samples. Pathway enrichment analysis of these highly correlated genes (Pearson’s correlation ≥ 0.674, *p* ≤ 8.07 × 10^−19^) highlighted functional processes of strong relevance for PDAC (Figure 1D), such as transcription and cell cycle regulation, amino acid metabolism and mitophagy/autophagy [2,25]. Together, these data support the notion that DDX21 is part of a pro-oncogenic program in PDAC.

### 3.2. DDX21 Is Upregulated in PDAC Liver Metastases

Poor prognosis in PDAC is associated with progression of the disease to the metastatic stage [1,2,3,25], with the liver being the most frequent distant site of seeding of metastatic cells. To assess whether *DDX21* expression was also associated with tumor progression in PDAC, we collected biopsies from matched primary lesions and liver metastases of 11 patients who underwent EUS-TA for diagnostic purposes. The selected patients were prevalently males (*n* = 8), and their age ranged between 63 and 84 (Figure 2A). After RNA extraction [8,19], samples were analyzed by quantitative real-time PCR (qPCR) for *DDX21* expression, using the ribosomal *L34* transcript as reference [8]. This analysis indicated that *DDX21* is frequently upregulated in most metastatic samples (*p* = 0.05), with eight cases showing increased expression in metastases (*p* = 0.006) and only three cases (two males and one female) showing slightly reduced expression (Figure 2B). Moreover, by querying expression data from TCGA PDAC samples, we observed a significant correlation between the expression of *DDX21* and that of genes known to be involved in PDAC metastasis (Figure 2C), such as Zinc Finger E-Box Binding Homeobox 1 (*ZEB1*) [26], Yes 1 Associated Transcriptional Regulator (*YAP1*) [27] and *QKI* [8].

Together, these data suggest the possible contribution of DDX21 in the progression of the disease to a metastatic stage.

### 3.3. DDX21 Is Equally Expressed in Classic and Basal-like PDAC

Mounting evidence suggests that classical PDAC displays a more favorable prognosis with respect to basal-like PDAC, in part due to the improved response to the multidrug therapy mFOLFIRINOX [2,4]. Thus, we asked whether the expression of *DDX21* was associated with a specific PDAC molecular subtype. However, computational analysis of transcriptomic data from TCGA showed equal expression of *DDX21* in samples that were classified as classical and basal-like based on Moffit’s gene signature [28] (Figure 3A). Coherently, qPCR (Figure 3B) and western blot (Figure 3C,D) analyses showed no clear difference in *DDX21* mRNA and protein expression between PDAC cell lines that are representative of the classical (HPAF-II, Capan-1), intermediate (AsPC-1) or basal-like (PANC-1, MiaPaCa-2) subtypes [8,22,29]. Indeed, while basal-like cells tended to have a higher expression of DDX21 than Capan-1 and AsPC-1 cells, the classical HPAF-II cell line also expressed high levels of this helicase. Together, these data suggest that DDX21 is equally expressed in both molecular subtypes of PDAC.

DDX21 was reported to play multiple roles in RNA metabolism, from transcription to transcript stabilization [30]. To evaluate the subcellular localization of DDX21 in PDAC cells, we carried out an immunofluorescence analysis of the endogenous protein in HPAF-II, MiaPaCa-2 and PANC-1 cells.

In all the cells tested, DDX21 was prevalently localized in nucleoli, with part of the protein being also diffused in the nucleoplasm (Figure 3E). By contrast, no cytoplasmic staining was observed. Thus, DDX21 is likely involved in nuclear RNA processing in PDAC cells.

### 3.4. Knockdown of DDX21 Exerts Widespread Effects on the Transcriptome of PANC-1 Cells

To investigate the impact of DDX21 on the transcriptome of PDAC cells, we knocked down its expression in PANC-1 cells and performed high-throughput (>100 million reads/sample) RNA-seq analyses [8,20,21]. DDX21 protein expression was efficiently downregulated after transfection of cells with specific siRNAs (siDDX21) with respect to control siRNAs (siCTRL; Figure 4A). DDX21 depletion resulted in differential expression of 760 genes, with 62.9% of them being upregulated and 37.1% downregulated (Figure 4B and Appendix A). In addition, we identified 476 differentially regulated alternative splicing events in 414 genes (Figure 4C and Appendix A). Alternative First Exon (AFE; 17.4%) and Exon Cassette (EC; 23.7%) were the predominant splicing patterns affected by DDX21 depletion (Figure 4C). Notably, ~50% of the regulated splice variants are not annotated as alternative splicing events (Figure 4C), hence deriving from regulation of both exons (21%) and introns (29.6%) that are annotated as constitutive in the reference FAST-DB database [8,20,21]. The KEGG analysis of genes upregulated in DDX21-depleted cells highlighted a significant enrichment for terms related to several pathways with relevance for PDAC, such as mitophagy/autophagy [31,32], necroptosis [33] and WNT pathway [34] (Figure 4D). Downregulated genes were enriched in pathways in cancer, Hippo signaling and transcriptional misregulation in cancer (Figure 4E). Conversely, splicing-regulated genes were enriched in terms that are not strictly related to PDAC (Figure 4F). Analysis by qPCR using an independent set of PANC-1 cells validated the differential expression of genes associated with autophagy, like the GABA Type A Receptor Associated Protein Like 1 (*GABARAP1L*) and the Mitogen-Activated Protein Kinase 8 (*MAPK8*), necroptosis, like the Poly(ADP-Ribose) Polymerase 1 (*PARP1*), pathways in cancer, like *CCND3*, *CDK6* and the CXC Motif Chemokine Ligand 2 (*CXCL2*) and Hippo signaling, like *YAP1* (Figure 4G–J). These data suggest that the impact exerted by DDX21 on the transcriptome of PDAC cells likely affects their biological features.

### 3.5. DDX21 Depletion Enhances Autophagy in PANC-1 Cells

Our transcriptomics analysis indicated that genes associated with mitophagy and autophagy are upregulated in cells depleted of DDX21 (Figure 4D,G,I). Since autophagy is involved in several steps of pancreatic tumorigenesis [32], we set out to further validate its regulation by DDX21. Western blot analysis of Autophagy Related 5 (ATG5) and lipidated Microtubule Associated Protein 1 Light Chain 3 Alpha (LC3), two proteins required for autophagosome formation, indicated their increase in PANC-1 cells depleted of DDX21 (Figure 5A,B). To test whether the autophagic flux was increased in DDX21-depleted PANC-1 cells, we treated them for one hour with Bafilomycin A1 to block the fusion of autophagosomes with lysosomes [35] and analyzed LC3 distribution by immunofluorescence. Coherently, with an increase in autophagy, cells lacking DDX21 displayed a larger accumulation of LC3-positive autophagosomes upon block of the autophagic flux (Figure 5C,D). Collectively, these data indicate that a decreased expression of DDX21 enhances the basal level of autophagy in PDAC cells.

### 3.6. DDX21 Promotes the G1-S Cell Cycle Progression and Proliferation of PANC-1 Cells

Among the genes downregulated in DDX21-depleted cells, we noted *CCND3* and *CDK6*. These genes encode for the cyclin D-CDK6 complex that is required for the cell cycle transition from G1 to S phase and represents an actionable vulnerability in several cancer types [36]. Notably, cyclin D3 was identified as the main D-type cyclin involved in PDAC cell proliferation, whose depletion led to the reduced expression of cell-cycle-associated genes and inhibition of cell cycle progression [37]. First, we confirmed that DDX21 knockdown significantly reduced also the protein levels of both cyclin D3 and CDK6 (Figure 6A). In line with the well-established role of the cyclin D-CDK6 complex [36], FACS analysis showed that knockdown of DDX21 caused a significant enrichment in G1 and reduction in S phase cells (Figure 6B,C). Furthermore, DDX21 depletion significantly reduced PANC-1 cell proliferation and clonogenic potential (Figure 6D,E). However, no impact of DDX21 depletion was observed on the sensitivity of PANC-1 cells to treatment with mFOLFIRINOX and Gemcitabine (Figure 6F,G). Since these drugs mainly target cells in the S and G2/M phase of the cycle, the lack of effect is consistent with the accumulation of DDX21-depleted cells in the G1 phase and their reduced proliferation (Figure 6B–D). These observations indicate that the reduction in expression of cyclin D3 and CDK6 elicited by DDX21 depletion has a functional impact on the proliferation potential of PDAC cells, without affecting their sensitivity to chemotherapy.

## 4. Discussion

DDX21 is a ubiquitous DEAD box RNA helicase that is highly expressed in proliferating tissues and plays a multilayered role in gene expression regulation [30]. Consistent with its main localization in nucleoli, DDX21 is deeply involved in the control of ribosomal RNA metabolism, by associating with RNA polymerase I (RNAPI) at ribosomal DNA locus and promoting the expression and processing of ribosomal RNA [10]. However, DDX21 can also localize to the nucleoplasm where it is recruited to genes transcribed by the RNAPII, facilitating the release of the enzyme from pausing [10]. Interestingly, the switch from nucleoli to nucleoplasm can be dynamically regulated by exogenous cues, such as the amount of glucose in the surrounding environment, with widespread consequences on transcriptional and post-transcriptional regulation of gene expression [13]. Moreover, DDX21 was shown to modulate the deposition of N6-methyladenosine (m6A) marks on nascent RNAs by interacting with the Methyltransferase 3 (METTL3) enzyme. This interaction was promoted by recognition of R loops by DDX21, while suppression of this interaction or inhibition of its helicase activity enhanced genomic instability [38]. Nevertheless, how these multiple functions of DDX21 are involved in tumorigenesis is still largely unknown. Indeed, the role of DDX21 in human cancers seems to be highly context-specific, with examples of both pro-tumoral and tumor-suppressive actions being reported [14,15,16,17,18]. In the context of PDAC, higher DDX21 expression was shown to be oncogenic and associated with worse prognosis [16,18]. In one recent study published while this work was in progress, DDX21 knockdown was shown to reduce proliferation, migration and invasion of MiaPaCa-2 cells [18], but the mechanisms or pathways involved were not investigated. Our study further supports the oncogenic role of DDX21 in PDAC and provides new insights into the genes and molecular processes affected by this helicase in PDAC cells.

*DDX21* expression was found to be elevated in PDAC with respect to surrounding non-tumoral pancreatic tissue [18]. We now extend this observation and show that *DDX21* expression is also increased in liver metastases with respect to primary PDAC lesions. Importantly, we analyzed matched primary and metastatic lesions from the same patients undergoing EUS-TA for diagnostic purposes. This finding supports a role for DDX21 in the progression of the disease at an individual level. In line with this pro-metastatic function, we also found a significant correlation of the expression of DDX21 and that of factors that were shown to promote metastatic diffusion of PDAC, such as ZEB1 [26], YAP1 [27] and the splicing factor QKI [8,39]. A role for DDX21 in metastasis is also supported by the pro-invasive action elicited by this helicase in MiaPaCa-2 cells [18], a mesenchymal cell line that displays traits of the aggressive basal-like PDAC subtype [8,29]. Nevertheless, we did not observe a subtype-specific expression of DDX21, which was equally expressed also in classical cell lines and PDAC samples. Notably, a limitation of our study is that we used only cell lines derived from primary tumors, and not metastases. Thus, further studies employing metastatic cell lines and PDAC in vivo models, as well as DDX21-overexpressing cells, are necessary to fully elucidate the contribution of DDX21 to disease progression and metastatic spread of PDAC cells.

Our RNA-seq analysis uncovered a widespread effect of DDX21 depletion on the transcriptome of PANC-1 cells, with hundreds of genes regulated at either expression or splicing level. Genes that were either induced or repressed by DDX21 knockdown were of particular interest for PDAC biology. For instance, among the upregulated genes, we observed a significant enrichment in the mitophagy/autophagy term. Accordingly, western blot and immunofluorescence analyses confirmed enhanced basal autophagy in DDX21-depleted cells. Autophagy is an important regulator of PDAC tumorigenesis, which differentially contributes to onset and progression of the disease [32]. At onset, autophagy plays a tumor-suppressive role by helping cells maintain genomic integrity, while inhibiting tissue damage and inflammation. At later stages, however, increased autophagy supplies nutrients to cancer cells and promotes their immune evasion through the degradation of MHC-I molecules on the plasma membrane [32]. Autophagy is often induced by nutrient deprivation, which can be mimicked by the inhibition of the mechanistic target of rapamycin complex 1 (mTORC1) [40]. Notably, mTORC1 is a major regulator of protein synthesis in the cell, which acts by promoting the assembly of the translation initiation complex and the synthesis of ribosomal proteins [41]. Thus, given the well-known role of DDX21 in ribosome metabolism [10], it is conceivable that the induction of autophagy is part of a compensatory feedback loop triggered by altered ribosome biogenesis in DDX21-depleted cells.

Our bioinformatics analysis also predicted a positive role for DDX21 in supporting several pathways of relevance for cancer. Among the genes associated with this ontology term and downregulated in DDX21-depleted PANC-1 cells, we observed genes associated with cell invasion, such as Matrix Metallopeptidase 1 (*MMP1*), or EMT, such as Transforming Growth Factor Beta Receptor 2 (*TGFBR2*), which are consistent with the proposed role of DDX21 in invasion [18]. Herein, we focused on *CCND3* and *CDK6* for their pivotal role in cell cycle regulation [36]. The activity of the cyclin D-CDK6 complex dictates the progression to the S phase of the cell cycle. Coherently, we observed that the reduced expression of both these proteins in DDX21-depleted PANC-1 cells was associated with a significant reduction of cells in the S phase and an increase in the G1 phase. As expected from this cell cycle perturbation, both the proliferation rate and the ability to form colonies were significantly impaired upon the knockdown of DDX21. Thus, the gene expression changes observed in our transcriptomic analyses appear to be functionally relevant. Of note, it was recently shown that combined treatment with the CDK4/6 inhibitor palbociclib and the autophagy inhibitor hydroxychloroquine exerted a synergic effect on preclinical models of PDAC [42]. It will be interesting to evaluate whether these drugs also affect the expression or function of DDX21, which we found to regulate both these targets in PDAC cells.

## 5. Conclusions

Our study further supports the oncogenic function of DDX21 in PDAC cells and provides new insights into the global gene expression changes that accompany depletion of this helicase in PDAC cells. Furthermore, we uncover a new link between DDX21 function and autophagy regulation, which could be also exploited in new combined treatments for this currently uncurable disease.

## Figures and Tables

**Figure 1 cancers-17-00570-f001:**
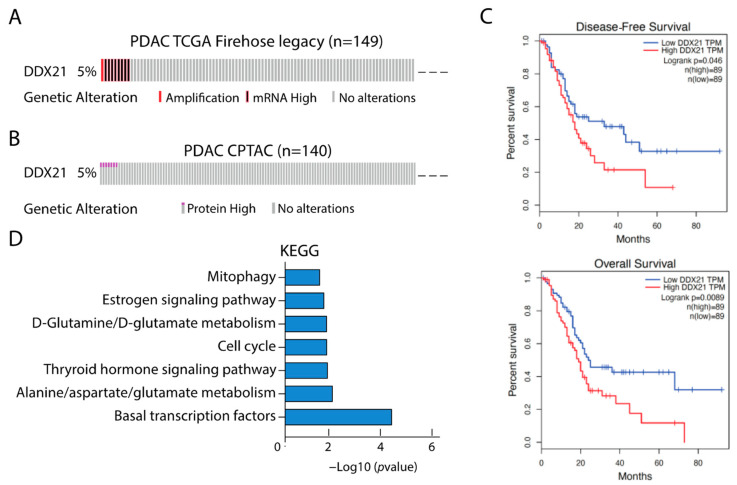
DDX21 has a prognostic value in PDAC. (**A**) Oncoplot showing the percentage of *DDX21* gene amplification or high expression in TCGA samples. (**B**) Oncoplot showing the expression level of DDX21 protein in CPTAC samples. (**C**) Kaplan–Meier curve displaying overall survival (**up**) and disease-free survival (**bottom**) of TCGA patients exhibiting high (red line) or low (blue line) expression of *DDX21*. (**D**) KEGG analysis of 100 genes with the highest positive expression correlation with *DDX21*.

**Figure 2 cancers-17-00570-f002:**
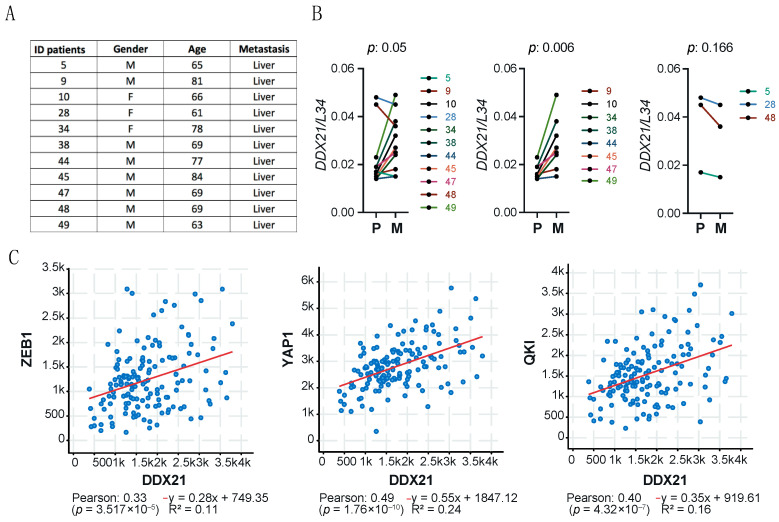
DDX21 expression in a cohort of PDAC patients. (**A**) List of the age, sex and metastatic site of the patients analyzed. (**B**) *DDX21* expression levels in primary lesion (P) and liver metastasis (M) from the eleven patients listed in (**A**). Statistical analysis was performed using the paired Student’s *t*-test. (**C**) Correlation analysis of the expression of *DDX21* and *ZEB1*, *YAP1* and *QKI* in TCGA PDAC samples.

**Figure 3 cancers-17-00570-f003:**
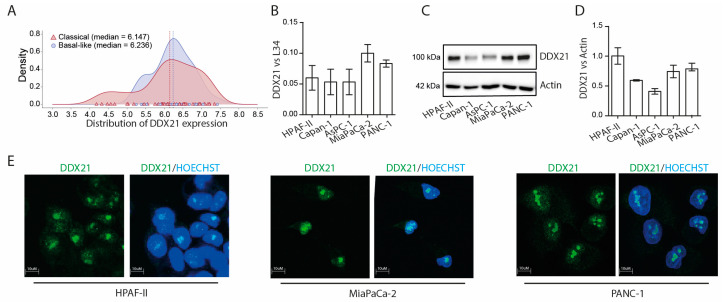
DDX21 expression and localization in PDAC cell lines. (**A**) Density plot of *DDX21* expression in classical (red) and basal-like (blue) PDAC samples from TCGA. (**B**) Quantitative real-time PCR analysis of *DDX21* expression in PDAC cell lines (HPAF-II, Capan-1, AsPC-1, MiaPaCa-2 and PANC-1) (*n* = 3). (**C**,**D**) Western blot analysis of DDX21 protein and histogram showing its quantification in PDAC cells (*n* = 3). (**E**) Immunofluorescence analysis of DDX21 subcellular localization in HPAF-II, MiaPaCa-2 and PANC-1 (*n* = 3).

**Figure 4 cancers-17-00570-f004:**
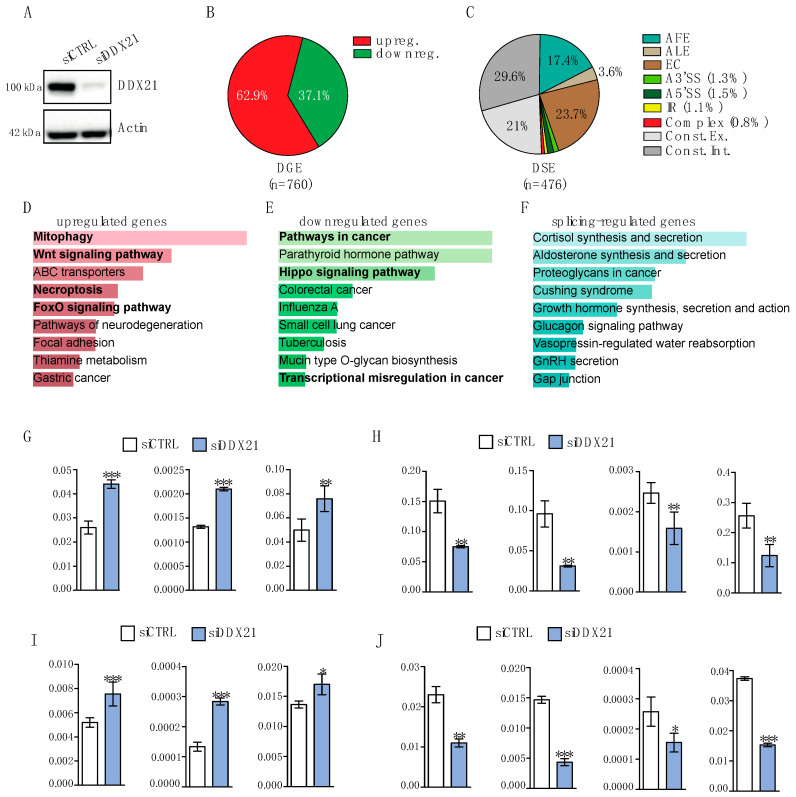
Transcriptome analysis of PANC-1 cells depleted of DDX21. (**A**) Western blot analysis of DDX21 in PANC-1 cells transfected with the corresponding small interfering RNAs (siRNAs) for 48 h. (**B**) Pie chart showing the percentage of upregulated (red) and downregulated (green) genes in DDX21-depleted cells. (**C**) Pie chart showing percentage of the indicated splicing patterns regulated by depletion of DDX21. AFE = alternative first exon, ALE = alternative last exon, A3’SS = alternative 3′ splice site, A5’SS = alternative 5′ splice site, IR = intron retention, Const. Ex. = constitutive exons, Const. Int. = constitutive introns. (**D**–**F**) KEGG analysis of upregulated genes (**D**), downregulated genes (**E**) and splicing-regulated genes (**F**) in DDX21-depleted cells. (**G**–**J**) qRT-PCR of selected upregulated (**G**,**I**) and downregulated (**H**,**J**) genes in DDX21-depleted cells (*n* = 3). *L34* (**G**,**H**) and *GAPDH* (**I**,**J**) were used as housekeeping genes for normalization of the expression data. Statistical analysis was performed with Student’s *t*-test (* *p* < 0.05, ** *p*< 0.01; *** *p* < 0.001).

**Figure 5 cancers-17-00570-f005:**
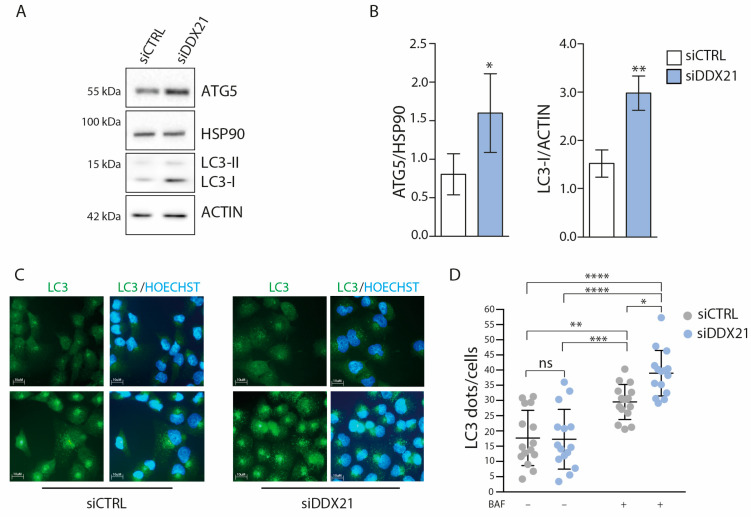
DDX21 regulates autophagy in PANC-1 cells. (**A**,**B**) Western blot analysis (**A**) and densitometric quantification (**B**) of ATG5 and LC3 levels in DDX21-depleted PANC-1 cells transfected with the corresponding small interfering RNAs (siRNAs). HSP90 and ACTIN were used as loading control for, respectively, ATG5 and LC3 (*n* = 3). Statistical analysis was performed with the Student *t*-test. (**C**) Representative immunofluorescence images showing LC3 staining in control and DDX21-depleted PANC-1 cells treated or not with Bafilomicin A1 (BAF) for 2 h (*n* = 3). (**D**) Quantification of the LC3 dots per cell was performed using the ImageJ quantification tool. Quantification was performed from 3 independent experiments, with 50 cells quantified for each individual sample. Error bars represent standard deviation (*n* = 3). Statistical analysis was performed by the Two-Way ANOVA (* *p* < 0.05, ** *p* < 0.01; *** *p* < 0.001; **** *p* < 0.0001).

**Figure 6 cancers-17-00570-f006:**
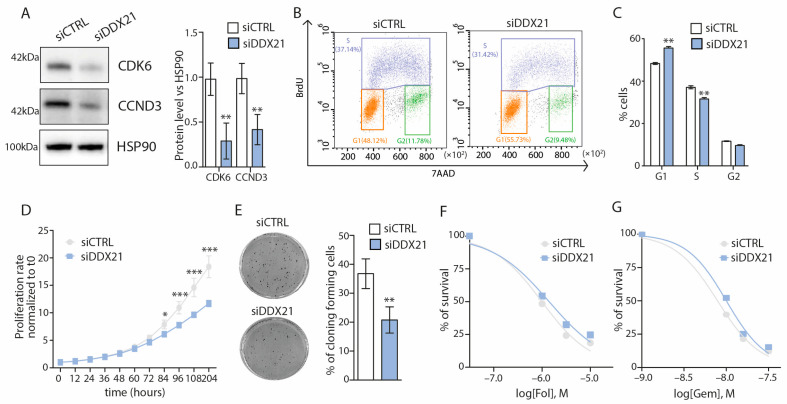
DDX21 promotes the G1-S transition and proliferation of PANC-1 cells. (**A**) Western blot analysis (**left**) and densitometric quantification (**right**) of CDK6 and cyclin D3 (CCND3) protein levels in DDX21-depleted PANC-1 cells. HSP90 was used as loading control (*n* = 3). Statistical analyses were performed by Student’s *t*-test (** *p* < 0.01). (**B**) FACS analysis showing DNA content (7AAD) and bromodeoxyuridine (BrdU) incorporation of PANC-1 cells transfected with ctrl or DDX21-targeting siRNAs for 48 h (*n* = 3). (**C**) The percentage of cells in G1, S and G2 phases are indicated (*n* = 3). Statistical analyses were performed by Student’s *t*-test (** *p* < 0.01). (**D**) Line graphs showing the proliferation rate of PANC-1 cells silenced for DDX21 or ctrl, evaluated as cell count normalized to T0 (*n* = 3). Statistical analyses were performed by the Two-Way ANOVA (* *p* < 0.05, *** *p* < 0.001). (**E**) Clonogenic assay of PANC-1 cells silenced or not for DDX21 after 10 days (*n* = 3). The histogram reports the percentage of seeded cells that formed colonies (Student’s *t* test; ** *p* < 0.01). (**F**,**G**) Line graphs showing the cytotoxic effect of mFOLFIRINOX (**F**) and Gemcitabine (**G**) in PANC-1 cells silenced (siDDX21) or not (siCTRL) for DDX21. Cells were exposed to the indicated concentrations of the drugs for 96 h and the viability was evaluated as cell count normalized to T0 (*n* = 3).

## Data Availability

The RNA-seq data are provided in the Appendix A.

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
