# Peer review of "DDX21 Controls Cell Cycle Progression and Autophagy in Pancreatic Cancer Cells"

_cancers, 2025, doi:10.3390/cancers17040570_

Round 1
Reviewer 1 Report
Comments and Suggestions for Authors
The manuscript characterizes the role DDX21 in PDAC. The work is coherently written and the logic of the experiments is clearly explained.
I have minor comments:
i) The authors point out that DDX21 is involved in RNA synthesis and splicing, particularly of ribosomal genes. I wonder why they decided to use L34 as housekeeping gene, given that it is a ribosomal gene. Authors should use a different type of housekeeping gene or demonstrate that L34 expression is not impacted by DDX21 expression. For example, is L34 impacted by DDX21 knock down? Does L34 expression correlate with any of the groups in the TCGA analysis done based on DDX21 expression?
ii) It would be important to control how the expression of top 100 genes used in Figure 1D change in the RNAseq performed.
iii) The authors suggest with this manuscript that DDX21 is particularly relevant in the context of liver metastasis. However, the cell lines used are derived from primary tumour. I would suggest to add a limitation to the study section in which they explain this and the fact that they did not perform overexpression experiments to corroborate their data.
iv) Methods section should be more specific. For example: section 2.3 on RNA analysis does not mention relevant details like sequencing depth (which is only mentioned in lines 234-235).
v) Methods section should have a statistic sub-paragraph. Moreover, statistic should be better explained in the legends. For example, in Figure 2B, was the student’s t test performed with one or two tails? And in Figure 4G and H, which statistical test was used?
vi) It should also be clearly written how many times each experiment was performed.
vii) The referencing to figures should be checked and improved. For example, it is missing the reference to Figure 2C in the text.
Author Response
Reviewer 1:
Comments and Suggestions for Authors
The manuscript characterizes the role DDX21 in PDAC. The work is coherently written and the logic of the experiments is clearly explained.
I have minor comments:
- i)The authors point out that DDX21 is involved in RNA synthesis and splicing, particularly of ribosomal genes. I wonder why they decided to use L34 as housekeeping gene, given that it is a ribosomal gene. Authors should use a different type of housekeeping gene or demonstrate that L34 expression is not impacted by DDX21 expression. For example, is L34 impacted by DDX21 knock down? Does L34 expression correlate with any of the groups in the TCGA analysis done based on DDX21 expression?
L34 expression was not significantly modulated by DDX21 knockdown in our RNA-seq dataset, and its expression was not among the top DDX21-correlating genes in the TCGA dataset. For these reasons, we used it to normalize the qPCR analyses. Nevertheless, as suggested by the Reviewer, we have now normalized the gene expression data with respect to another housekeeping gene (GAPDH). The qPCR results are shown in the new Figure 4I,J and are in line with those obtained by normalizing for L34.
- ii)It would be important to control how the expression of top 100 genes used in Figure 1D change in the RNAseq performed.
As suggested, we have performed an overlap of the DDX21 regulated genes and the top 100 DDX21-correlating genes. We found only 4 genes overlapping in the two datasets (TFAM, MAPK8, AFF4, POMK). This result suggests that DDX21 is part of a gene signature in PDAC but is not the determinant of such signature. However, it is also important to note that our RNA-seq data refer to the impact of DDX21 in a specific PDAC cell line (PANC-1), whereas the TCGA data refer to tumor biopsies that contain a heterogeneous population of tumor, stromal and immune cells.
iii) The authors suggest with this manuscript that DDX21 is particularly relevant in the context of liver metastasis. However, the cell lines used are derived from primary tumour. I would suggest to add a limitation to the study section in which they explain this and the fact that they did not perform overexpression experiments to corroborate their data.
We agree with the Reviewer that this aspect represents a limitation of our study. We have now added a comment explaining this issue in the revised Discussion section (see lines 393-397).
- iv)Methods section should be more specific. For example: section 2.3 on RNA analysis does not mention relevant details like sequencing depth (which is only mentioned in lines 234-235).
We have now added the requested information in paragraph 2.3 of the Methods section.
- v)Methods section should have a statistic sub-paragraph. Moreover, statistic should be better explained in the legends. For example, in Figure 2B, was the student’s t test performed with one or two tails? And in Figure 4G and H, which statistical test was used?
We have now added a paragraph (2.7) with description of the statistical analyses used in our study in the revised Methods section and figure legends, when appropriate.
- vi)It should also be clearly written how many times each experiment was performed.
We apologize for this mistake. We have now stated the number of experiments performed in the revised figure legends or in the Methods section.
vii) The referencing to figures should be checked and improved. For example, it is missing the reference to Figure 2C in the text.
We have now corrected this mistake in the revised text.
Reviewer 2 Report
Comments and Suggestions for Authors
Cancers-3417088
Type of manuscript: Article
Title: DDX21 controls cell cycle progression and autophagy in pancreatic cancer cells
Authors: Adriana Leccese, Veronica Ruta, Valentina Panzeri, Fabia Attili, Cristiano Spada, Valentina Cianfanelli, Claudio Sette *
This study provides valuable insights into the role of DDX21 in PDAC, highlighting its potential as a therapeutic target. Further research to address mechanistic details, in vivo validation, and clinical translation could strengthen the study's impact. However, there are several key issues. Please review the points mentioned below and make the necessary corrections.
[Major concerns]
1. Institutional Review Board (IRB) Statement: This study mentions the use of tissue samples from patients, but there is no description of IRB approval anywhere in the text. Specifically, the IRB Statement at the end of the submission guidelines for the journal 'Cancers' has been omitted. If IRB approval was indeed not obtained, the paper should be rejected regardless of the review results, or it should be rejected now. In any case, a clear determination regarding IRB approval must be made first.
2. Detailed transcriptomics analysis: Highlight key genes or pathways affected by DDX21 depletion and their relevance to PDAC biology.
3. Combination therapies: Explore whether targeting DDX21 enhances the efficacy of existing chemotherapeutic agents.
4. Nomenclatures of cell lines: The study uses five types of cells, but their names are inconsistent. Please correct this properly. Examples: Capan-1 vs. CAPAN-1; AsPC-1 vs. AsPC1; MiaPaCa-2 vs. MiaPaCa2; etc. Additionally, please consistently list these cells in the same order as initially mentioned, and maintain this order in the figures or tables presenting the experimental results. This will help enhance the readers' understanding. Examples: Lines 208 and 215, etc.
5. Abbreviations: The use of abbreviations when writing a paper has many advantages besides simplicity of expression. To use an abbreviation, first write the abbreviation in parentheses after the full name, and then use the abbreviation from Introduction to the final Conclusion. Abbreviations should only be used if they are repeatedly used and if they are not used again, only the full name should be used.
6. In cases where abbreviations are used within figures or tables, please list these abbreviations along with their corresponding full names in the figure legends or at the bottom of corresponding tables. If there are two or more abbreviations, arrange them in alphabetical order. In this case, non-proper nouns should not have their first letters capitalized.
7. Materials and Methods section - When naming a particular chemical company, you must provide location information such as company name, city and/or state (abbreviation in the USA and Canada) and country. Once you have named a company with the information, you should only mention a company’s name thereafter.
8. Notation of units: Ensure consistent notation for units when referring to the same measurement. Examples: ‘h’ vs. ‘hr’ vs. ‘hour’ vs. ‘hours’. In addition, in scientific papers, almost all units, except for temperature and percentage, are written with a space between the number and the unit. However, in this paper, numbers and units are written together without spaces in both the figures and the text, which is very distracting. All units should be written with a space. Examples: ‘kDa’.
9. English: The English manuscript is generally well-written; however, there are instances where the names of certain human diseases, chemicals, and proteins are unnecessarily capitalized. Please review the entire manuscript and correct all such instances.
10. Statistical analysis: If statistical analyses were performed on the experimental results, please include the statistical outcomes in the figures as well. As far as I know, statistically, *p<0.05, **<0.01, and ***p<0.001 are enough. In addition, statistical annotations should only be included in the figure legends for the respective figures. Unnecessary statistical annotations can be omitted. Delete **** at Fig. 5D.
[Minor concerns]
1. Line 20: Pancreatic Ductal Adenocarcinoma should be written as Pancreatic ductal adenocarcinoma.
2. Line 98: The speed of the centrifuge should be expressed in gravity (g) rather than rpm. When indicating centrifugal force, either italicize the 'g' representing gravity or write it as 'x g'.
3. Line 189: Nomenclatures of human genes: Many human genes are mentioned in the paper, and while some are italicized, many others are not. Please review the manuscript thoroughly and ensure all gene names are correctly italicized. In addition to these, there are many incorrect human gene notations. Please find and correct them all. Examples: Line 189; Lines 289 and 290; etc.
4. Line 125: CO2 should be written as CO2.
5. Line 135: Protein tech should be written as Protein Tech.
6. Lines 153 and 154: Re-write ‘(0.125 lg 9 sample)’ and ‘(1 lgmL1)’.
7. Line 298: Re-write (fig. 6B,C).
8. Line 308: *p<0.05, not *p<0.005.
[Final conclusion]
1. If IRB approval for this study was not obtained, I recommend rejecting the publication.
2. If IRB approval was obtained and it will be included in the text during the revision stage, I recommend a major revision.

Author Response
Reviewer 2:
Comments and Suggestions for Authors
This study provides valuable insights into the role of DDX21 in PDAC, highlighting its potential as a therapeutic target. Further research to address mechanistic details, in vivo validation, and clinical translation could strengthen the study's impact. However, there are several key issues. Please review the points mentioned below and make the necessary corrections.
[Major concerns]
- Institutional Review Board (IRB) Statement: This study mentions the use of tissue samples from patients, but there is no description of IRB approval anywhere in the text. Specifically, the IRB Statement at the end of the submission guidelines for the journal 'Cancers' has been omitted. If IRB approval was indeed not obtained, the paper should be rejected regardless of the review results, or it should be rejected now. In any case, a clear determination regarding IRB approval must be made first.
We apologize for this mistake. The study was approved by the Ethical Committee and Institutional Review Board before its beginning. We have now added this information and the protocol ID in the revised paragraph 2.1, (line 94-99) of the Methods section.
- Detailed transcriptomics analysis: Highlight key genes or pathways affected by DDX21 depletion and their relevance to PDAC biology.
In the text we highlighted the genes associated with the phenotype that we described (autophagy, cell cycle transition). We now also highlight two genes, MMP1 and TGFBR2, that are involved in cell invasion and EMT, two processes associated with metastases (see revised Discussion, lines 430-433. A complete list of the genes regulated by DDX21 at expression and splicing level is provided in the Supplementary Tables S2 and S3.
- Combination therapies: Explore whether targeting DDX21 enhances the efficacy of existing chemotherapeutic agents.
To address this request of the Reviewer, we have now tested the sensitivity of DDX21-depleted PANC-1 cells to mFOLFIRINOX and Gemcitabine, the two chemotherapeutic treatments generally used as first-line therapy for PDAC. As shown in the new Figure 6F,G, we did not observe significant changes in the sensitivity of DDX21-depleted PANC-1 cells to these treatments. Since these drugs mainly target cells in the S and G2/M phase of the cycle, the lack of effect is consistent with the accumulation of DDX21-depleted cells in the G1 phase and their reduced proliferation (Fig. 6B-D). We discussed these data in the revised text (see lines 344-348).
- Nomenclatures of cell lines: The study uses five types of cells, but their names are inconsistent. Please correct this properly. Examples: Capan-1 vs. CAPAN-1; AsPC-1 vs. AsPC1; MiaPaCa-2 vs. MiaPaCa2; etc. Additionally, please consistently list these cells in the same order as initially mentioned, and maintain this order in the figures or tables presenting the experimental results. This will help enhance the readers' understanding. Examples: Lines 208 and 215, etc.
We have modified the nomenclatures of cell lines to maintain consistency. We also maintained the same order of cell lines in the panels of Figure 3, where applicable.
- Abbreviations: The use of abbreviations when writing a paper has many advantages besides simplicity of expression. To use an abbreviation, first write the abbreviation in parentheses after the full name, and then use the abbreviation from Introduction to the final Conclusion. Abbreviations should only be used if they are repeatedly used and if they are not used again, only the full name should be used.
We have followed the indication of the Reviewer in the revised text. The only exception is for the name of genes, which have been abbreviated with their acronym even if used only once.
- In cases where abbreviations are used within figures or tables, please list these abbreviations along with their corresponding full names in the figure legends or at the bottom of corresponding tables. If there are two or more abbreviations, arrange them in alphabetical order. In this case, non-proper nouns should not have their first letters capitalized.
We have modified the abbreviations in the figure legend as requested.
- Materials and Methods section - When naming a particular chemical company, you must provide location information such as company name, city and/or state (abbreviation in the USA and Canada) and country. Once you have named a company with the information, you should only mention a company’s name thereafter.
We have added the requested information in Materials and Method section.
- Notation of units: Ensure consistent notation for units when referring to the same measurement. Examples: ‘h’ vs. ‘hr’ vs. ‘hour’ vs. ‘hours’. In addition, in scientific papers, almost all units, except for temperature and percentage, are written with a space between the number and the unit. However, in this paper, numbers and units are written together without spaces in both the figures and the text, which is very distracting. All units should be written with a space. Examples: ‘kDa’.
We have modified the text according to the indication of the Reviewer.
- English: The English manuscript is generally well-written; however, there are instances where the names of certain human diseases, chemicals, and proteins are unnecessarily capitalized. Please review the entire manuscript and correct all such instances.
We modified this aspect as suggested in the main text.
- Statistical analysis: If statistical analyses were performed on the experimental results, please include the statistical outcomes in the figures as well. As far as I know, statistically, *p<0.05, **<0.01, and ***p<0.001 are enough. In addition, statistical annotations should only be included in the figure legends for the respective figures. Unnecessary statistical annotations can be omitted. Delete **** at Fig. 5D.
We modified the statistical annotations as suggested in the figure legends.
[Minor concerns]
- Line 20: Pancreatic Ductal Adenocarcinoma should be written as Pancreatic ductal adenocarcinoma.
We have corrected this mistake.
- Line 98: The speed of the centrifuge should be expressed in gravity (g) rather than rpm. When indicating centrifugal force, either italicize the 'g' representing gravity or write it as 'x g'.
We have corrected this mistake.
- Line 189: Nomenclatures of human genes: Many human genes are mentioned in the paper, and while some are italicized, many others are not. Please review the manuscript thoroughly and ensure all gene names are correctly italicized. In addition to these, there are many incorrect human gene notations. Please find and correct them all. Examples: Line 189; Lines 289 and 290; etc.
We have corrected this mistake.
- Line 125: CO2 should be written as CO2.
We have corrected this mistake.
- Line 135: Protein tech should be written as Protein Tech.
We have corrected this mistake.
- Lines 153 and 154: Re-write ‘(0.125 lg 9 sample)’ and ‘(1 lgmL1)’.
We thank the Reviewer for the suggestion. We have now modified the main text in “Cell cycle was evaluated by flow cytometry using single staining with 7-aminoactinomycin D (7-AAD; 2.5 g/ml, Biotium, Fremont, CA, USA) or double staining with 7-AAD and anti-BrdU antibody (0.125 mg for each sample, in the presence of ribonuclease A (1 mg×mL-1) as previously described.”
- Line 298: Re-write (fig. 6B,C).
We have corrected this mistake.
- Line 308: *p<0.05, not *p<0.005.
We have corrected this mistake.
[Final conclusion]
- If IRB approval for this study was not obtained, I recommend rejecting the publication.
- If IRB approval was obtained and it will be included in the text during the revision stage, I recommend a major revision.
The protocol for this study was approved by the IRB. We have now added the protocol ID in the section 2.1 of the material and methods, lines 94-99.
Round 2
Reviewer 2 Report
Comments and Suggestions for Authors
The authors have appropriately responded to and addressed the points raised in the first review; however, additional technical corrections are still necessary.
- General names of compounds, proteins, etc., are not typically capitalized within sentences, but this issue has yet to be fully corrected.
- Information about reagent companies (company name, city, state, country) has been largely revised. However, if a company has already been mentioned once, subsequent references should include only the company name, omitting additional details such as city and country. Since the purpose is to provide information, repeating the same details is unnecessary.
- Many other technical corrections are still required, but I hope the Office will provide thorough support during the proofreading stage.
The authors have appropriately responded to and addressed the points raised in the first review; however, additional technical corrections are still necessary.
- General names of compounds, proteins, etc., are not typically capitalized within sentences, but this issue has yet to be fully corrected.
- Information about reagent companies (company name, city, state, country) has been largely revised. However, if a company has already been mentioned once, subsequent references should include only the company name, omitting additional details such as city and country. Since the purpose is to provide information, repeating the same details is unnecessary.
- Many other technical corrections are still required, but I hope the Office will provide thorough support during the proofreading stage.